# Perceived Benefits of Nature in Diverse Populations

**DOI:** 10.3390/ijerph22040563

**Published:** 2025-04-04

**Authors:** Joy L. Hart, Kandi L. Walker, Cameron K. Stopforth, Anna Simpson, Aruni Bhatnagar, Rachel J. Keith

**Affiliations:** 1Department of Communication, College of Arts and Sciences, University of Louisville, Louisville, KY 40292, USA; joy.hart@louisville.edu (J.L.H.); kandi.walker@louisville.edu (K.L.W.); 2Christina Lee Brown Envirome Institute, School of Medicine, Physiology Division, University of Louisville, Louisville, KY 40202, USA; cameron.stopforth@louisville.edu (C.K.S.); aruni.bhatnagar@louisville.edu (A.B.); 3School of Public Health, Drexel University, Philadelphia, PA 19104, USA

**Keywords:** demographic determinants, socioeconomic determinants, perceived benefits of nature

## Abstract

The relationship between socioeconomic and demographic characteristics and health effects of green spaces has been studied, suggesting that certain groups may reap more health benefits from exposure to nature. However, the link between the perceived benefits of nature and socioeconomic and demographic characteristics remains a gap in the literature. We used a subsample (*n* = 711, 2018–2019) from an environmental cardiovascular risk cohort to investigate the perceived benefits of nature. Participants completed an 11-item survey about their perceptions of the benefits of nature at in-person visits. Socioeconomic and demographic characteristics including income, education, race, biological sex at birth, and age, were self-reported. Generalized linear models were used to evaluate associations between the perceived benefits of nature and demographic and socioeconomic factors; odds ratios and 95% confidence intervals (CIs) are reported. Both unadjusted and fully adjusted models for race, age, sex, and education are reported. Our results suggest that participants who identified as male, a member of a minoritized population, and/or completing less education perceived nature as less beneficial. Although additional research is needed to better understand contributors to these perceptions, access to convenient, safe, and multi-use green spaces may be important in encouraging time in nature and shifting perceptions of the benefits of greenness.

## 1. Introduction

Considerable evidence points to the many benefits of living near and spending time in green spaces. Such benefits of exposure to nature encompass physiological, psychological, social, and environmental outcomes. In the physiological area, research using both cross-sectional and longitudinal data has reported positive associations between living in green areas and lower all-cause and cardiovascular mortality, as well as decreased risk for cardiovascular disease [1,2,3,4]. Similarly, positive associations have been documented across multiple studies between greenness exposure and lower respiratory mortality, better lung function, and decreased incidence of lung cancer. In addition, greenness exposure has been associated with fewer preterm, low weight, and small gestational-age births [5,6].

In the psychological arena, exposure to greenness (i.e., green spaces or areas with considerable vegetation, such as tree canopy or grass) has been positively linked with psychological well-being [7], mental health [4,8,9], cognitive function (e.g., restoration, attention, and working memory) [10,11,12], brain activity (e.g., regulation of emotions) [10,13,14] and mood [15,16]. Additionally, time in green spaces has been positively associated with reductions in thought rumination, stress, and anxiety [10], all of which may influence overall mental health and functioning.

In terms of social outcomes, several studies have found positive associations between exposure to nature and social health [17]. Areas of social health studied include social cohesion [18,19], social support [20], social relationship quality [21], and social connectedness [22]. Relatedly, these social dimensions can influence as well as be influenced by physical activity, another dimension of health studied in relation to green spaces [5,10,23].

In the area of environmental outcomes, green areas are believed to have the potential to mitigate some of the effects of climate change, such as shaping more comfortable temperatures, reducing flooding, and mitigating heat island effects [24,25,26,27,28]. Other outcomes included improved sleep through decreases in noise and light pollution [10,29,30,31], improving the human microbiome [32], and facilitating exposure to beneficial volatile organic compounds [2].

Although additional population and longitudinal studies are needed to further explicate these findings and examine the mechanisms underlying them, the results overall are promising for human health. Given factors associated with the built environment in urban areas, exposure to green spaces may be especially important to city dwellers and important in urban planning [9,33] with nature-based solutions, such as increased urban green spaces, a comparatively low-cost approach to addressing concerns and offering an array of positive health benefits. For example, exposure to green spaces has been shown to improve longevity and circulatory diseases in those who have the least economic resources [34]. Additionally, the elderly and children also tend to see more health benefits from green exposure [34].

Many benefits ascribed to various demographic and socioeconomic groups are derived from epidemiologic studies that do not assess thoughts about or perceptions of the greenness. Although some of the potential benefits are derived unobtrusively (e.g., trees filtering pollution from the air that humans breathe), others are influenced by human behavior (e.g., talking with neighbors, spending time in parks or other green spaces). Thus, it is important to understand how community members view the benefits of green spaces and time in nature, as their behavior will influence the resulting individual and community advantages. Studies that address this gap in the academic literature are needed to enhance both scientific understanding of the human-nature interface as well as urban planning implementations and policy development.

In short, individual beliefs shape interest in, actions taken regarding, engagement with, and overall support for green spaces. Some previous work has examined factors that influence the perceived benefits of nature, and such factors are especially important in understanding outcomes in urban greening initiatives (i.e., adding or enhancing green spaces in cities through actions such as green rooftops, living walls, and microforests). Therefore, this study sought to investigate the perceived benefits of nature based on demographic and socioeconomic factors among cohort members of a community-based clinical trial with an urban greening intervention. More specifically, we investigated two research questions: What are the perceived benefits of nature among cohort members enrolled in a community-based clinical trial with an urban greening intervention? How do cohort members’ perceptions vary based on demographic and socioeconomic factors?

## 2. Methods

All participants were enrolled in an ongoing controlled, community-based clinical trial evaluating the effect of increased greenness on community health in Louisville, Kentucky. All study-related procedures and measures were approved by the University of Louisville’s Institutional Review Board (IRB #15.126), and informed consent was obtained prior to collecting any data. Social, demographic, and perceptions of nature information were collected by self-reported questionnaires at in-person visits. The responses were collected and managed using Research Electronic Data Capture (REDCap 14.5.17) electronic data capture tools hosted at the University of Louisville [35,36]. A total of 730 participants completed wave 1 of the study, and for this analysis, participants with missing data on ethnicity (*n* = 2), income (*n* = 3), education (*n* = 4), and the Perceived Benefits of Nature Questionnaire (PBNQ; *n* = 10) were excluded, resulting in a final analytic sample of *n* = 711 (see Appendix A for study flow chart).

### 2.1. Participants

Participants were enrolled in the Health, Environment, and Action in Louisville (HEAL) study, part of the Green Heart Louisville project, from May to October in 2018 and 2019. The study design and details have been previously published [37]. Briefly, inclusion criteria included adults aged 25–70 years, residing within our 4-square mile study region, and free of certain chronic or acute diseases that could not be controlled for within our study design [37]. Each in-person study visit took approximately 2 h, allowing participants to self-report demographic information, medical status (physical and mental), psychosocial details, occupational information, and lifestyle choices, as well as complete a series of physical measures. Staff were on hand to complete the physical measures and assist with the collection of self-report data as needed. Compensation was provided for time.

### 2.2. Perceived Benefits of Nature

Individuals’ perceptions of how nature benefited their mental and physical health were assessed using the Perceived Benefits of Nature Questionnaire (PBNQ) [38], a reliable and validated measure for examining such perspectives. Briefly, this 11-item measure assesses whether anticipatory benefits from nature predict health benefits derived from nature exposure. Participants used a 7-point scale (i.e., 1 = “Extremely uncharacteristic of me”; 7 = “Extremely characteristic of me”) to respond to each of the 11 items. Example PBNQ items include “Interacting with nature keeps me optimistic”, “I feel relaxed when I think of nature”, and “In order to maintain physical health, I have to interact with nature”. Scores were summed, including appropriate reverse scoring of negative items, with a higher score indicating a greater belief in the benefits of nature [38].

### 2.3. Demographic Questionnaires

Self-reported socioeconomic and demographic information was collected using validated or harmonized data collection instruments available as recommended by NIH standards. Briefly, participants self-selected their racial identity (i.e., American Indian or Alaskan Native, Asian, Black or African American, Hawaiian or Pacific Islander, White, Other), ethnicity (Hispanic/Latino, and biologic sex (female, male, and intersex). Age was calculated from each participant’s birthdate, which was provided at the study visit. Income data were collected in the following brackets: <$20,000, $20,000–44,999, $45,000–64,999, $65,000–89,999, $90,000–124,999, and ≥$125,000 and divided into tertiles (i.e., low ≤ $45,000, medium = $45,000–90,000, and high ≥ $90,000). The response options for education completed were as follows: <high school; some high school; high school graduate or General Education Development test (GED); some college; 2-year degree or certificate; 4-year degree; Master’s; and Doctorate.

### 2.4. Statistical Analysis

Descriptive statistics were calculated for the full data set. For demographic data, frequencies were calculated for the categorical data and compared via the Chi-Square test (or Fisher’s Exact test if the expected cell count was less than 5). Race was dichotomized to White, and Other racial identities. Age was divided into 5 ranges (25–34, 35–44, 45–54, 55–64, and 65 or greater years), with each range spanning about a decade. Sex determined at birth was designated as male and female; no participants reported being intersex. Education completed was dichotomized to some college or less (i.e., <high school, some high school; high school graduate or GED; some college) and associates degree or higher (i.e., 2-year degree or certificate, 4-year degree, Master’s, Doctorate). Due to collinearity with education, income was not included in the statistical analysis. For linear regression, we categorized minority race groups together and used White race as the comparator group. Generalized linear models were used to evaluate associations between the perceived benefits of nature and demographic and socioeconomic factors; odds ratios and 95% confidence intervals (CIs) are reported. Both unadjusted and fully adjusted models for race, age, sex, and education are reported. The significance level for statistical tests was set to 0.05. Data were analyzed using R version 4.3.1 [39].

## 3. Results

The mean age of participants was 50 (±13) years, 61% reported being female, and 21% self-identified as a member of a minoritized population. Participants who were members of a minority population were more likely to be middle-aged and report low income and educational background (*p*-values < 0.001) (Table 1).

Mean perceived benefits of nature was calculated for each sociodemographic category (Appendix B). Mean scores ranged from 54 ± 14 in 45–54-year-old participants to 59 ± 12 in those aged 25–34 years. Interestingly, participants aged 65 years and older had the second highest perceptions of nature mean (58 ± 11). White participants scored higher than minoritized populations (57 ± 13 vs. 53 ± 15, respectively). Females perceived more benefits from nature than men (57 ± 14 vs. 54 ± 14). Higher income and education also resulted in higher sum scores than lower education and income categories.

In unadjusted univariable linear regression models, we saw significant negative associations of the perceived benefits of nature for member of minority population (β = −3.82 (−6.25, −1.39), *p* = 0.002), middle aged (β = −5.21 (−8.59, −1.84), *p* = 0.002), male sex (β = −3.41 (−5.50, −1.32), *p* = 0.001), and less education (β = −5.72 (−7.76, −3.68), *p* < 0.001) (Table 2). In fully adjusted models (i.e., adjustments for potential influence of race, age, sex, and education), we saw the same significant predictors of a negative association with perceived benefits of nature (All *p* < 0.05). The results after adjustment were as follows: being middle aged (β = −4.40 (−7.71, −1.08), *p* = 0.009), less educated (β = −4.92 (−6.98, −2.86), *p* < 0.001), male sex (β = −3.85 (−5.89, −1.81), *p* < 0.001), and a member of a minority population (β = −3.12 (−5.52, −0.71), *p* = 0.01). This model accounted for 7% of the variance seen in the perceived benefits of nature (Table 2).

## 4. Discussion

We examined associations between individuals’ perceived benefits of nature and demographic and socioeconomic factors in participants in the community-based HEAL clinical trial. Of the analytic sample, most were middle-aged (M_age_ = 50 years) and identified as female (61%), and approximately one-fifth (21%) identified as members of minoritized populations.

In adjusted models, participants who identified as male (vs. female), as a member of a minoritized population (vs. White), and as completing less education (vs. more education) perceived fewer benefits from nature. Previous research has indicated that perceptions of nature’s benefits vary and are influenced by social and individual factors [40]. The motivational theory, Maslow’s hierarchy of needs, supposes a five-tier model of human needs, with individuals first focused on the lower-level basic needs like food, water, and safety. After those needs are met, other needs, such as health, community, and love, can be addressed. Our findings that lower educated individuals who tend to have less income, those who are from minoritized populations have experienced historical environmental injustices, and women who often report more stress from caregiving roles align with this theory as described below.

### 4.1. Differences Based on Sex

According to previous work, men spend more time in nature [41], visit nature more often, and have reaped the health benefits of that proximity to greenness [42] compared with women. Hypotheses regarding factors fueling this gender gap include the time demands that typically influence women’s schedules more than men’s (e.g., caregiving for young children, other adults, and other family members and friends) and limit available opportunities to spend time in nature as well as concerns surrounding feeling vulnerable in nature and preferences to avoid such fears [43]. However, our results indicate that male participants did not perceive nature to be as beneficial as female participants despite the health benefits reported with exposure. Although additional inquiry is needed to further explore the factors influencing these different perceptions, perhaps women, who may have fewer opportunities to spend time in nature due to caregiving roles [44] and more concerns regarding personal vulnerability, long for time in green spaces and are more attuned to potential benefits. For example, research indicates that, compared to men, women appreciate features such as play areas for children and good lighting [45]. Additionally, a systematic review of green space exposures and physical health found that women benefitted more than men from greenness in close proximity to their residences, as well as overall greenness [46]. Future studies should explore multi-use areas (e.g., public parks) or residential greening, such as seen with the Green Heart Project, that may appeal to women and lessen safety concerns as well as social views that may influence individual opinions.

### 4.2. Differences Based on Race

Earlier research documents the under-representation of members of minoritized populations using green spaces [47] as well as differing perceptions of urban green spaces [48], and, most importantly, the frequent lack of green space in predominantly minority communities due to discriminatory decision-making in urban planning [49,50]. Given decades of structural racism, segregated housing, lack of economic investment in minority neighborhoods, and similar factors, it is not surprising that members of minoritized populations less frequently use community green spaces when they do exist, especially given frequent concerns with issues such as upkeep and environmental hazards [51]. The lack of availability and, when available, lack of use may serve to perpetuate health inequities and may shape perceptions of exposure to green space. For example, given the environmental injustices that many low-income and minority communities face, concerns may abound about time in nature and the possibility of benefits or harms. In our study, which encompassed low- to middle-income neighborhoods, members of minoritized populations did not perceive nature’s benefits as positively as did White participants. Although perhaps social norms influencing views of nature and health shaped these perceptions, additional research is needed to clarify the drivers of these differing viewpoints and to better understand necessary areas for change. An analysis of 44 cities where non-Hispanic Black citizens are the population majority indicated significant relationships between both increased tree canopy cover and overall greenness and lower prevalence of obesity [52]. However, much additional work remains to understand the perceptions that shape individuals’ engagement with green spaces.

### 4.3. Differences Based on Education

Numerous studies have linked higher educational attainment with greater overall health and longer lifespan [53], as well as more positive views toward the environment and nature. Previous work also has found that socioeconomic factors, such as income, influence perceptions of the benefits of nature [40] as well as available leisure time. In our analysis, participants with lower educational attainment (vs. higher educational attainment) perceived fewer benefits of nature. Although additional research is needed to further explore and explain this association, influences akin to those described above may be contributing factors. For example, perhaps convenient access to safe green spaces and open time to use such spaces wield influence. Additionally, future research should examine experience with nature-based education and incorporate variances across educational contexts, as perhaps individuals completing higher levels of education are more frequently engaged with such topics.

### 4.4. Limitations

Our findings should be considered in the context of the study’s limitations. One, the data are self-reported and thus subject to recall biases. Two, our sample is relatively small, which may have influenced the results. Three, the sample consists of participants enrolled in an ongoing health study in one geographic area; therefore, the results may not be generalizable to populations in other locations. Fourth, the data are cross-sectional. Future research would benefit from longitudinal data examining potential changes in perceptions across time. Five, we examined demographic and socioeconomic characteristics separately. Research with larger samples and an intersectional lens [47,54] would provide a more fine-grained assessment of group perceptions. Building from this work, future studies might follow cohorts in multiple and differing geographic locations across time, employing an intersectional approach to investigate group differences more fully, and urban planners and policymakers should assess differential access to safe, multipurpose green spaces across population groups.

## 5. Conclusions

This study examined associations between the perceived benefits of nature and demographic and socioeconomic factors in an urban adult cohort. Participants who identified as male, a member of a minoritized population, and/or completing less education perceived nature as less beneficial. Although additional research is needed to better understand contributors to these perceptions, future urban planning efforts should prioritize the creation and maintenance of accessible, convenient, safe, and multifunctional green spaces to enhance community perceptions of nature and its benefits for everyone, particularly for those who need it most. Such considerations are important in urban planning and policy development and may result in actions that encourage time in nature and shift perceptions of the benefits of greenness among sociodemographic groups.

## Figures and Tables

**Table 1 ijerph-22-00563-t001:** Characteristics of study participants.

Variables	Total (*n* = 711)	Minoritized Race (*n* = 162)	White Race (*n* = 549)
Age years (*n*, %)			
25–34	126 (18)	29 (18)	97 (18)
35–44	164 (23)	38 (24)	126 (23)
45–54	134 (19)	36 (22)	98 (18)
55–64	204 (29)	52 (32)	152 (28)
65 and over	83 (12)	7 (4)	76 (14)
Race (*n*, %)			
White	549 (77)	0 (0)	549 (100)
Black	127 (18)	127 (78)	0 (0)
Other	35 (5)	35 (22)	0 (0)
Sex (*n*, %)			
Female	434 (61)	110 (68)	324 (59)
Male	277 (39)	52 (32)	225 (41)
Household Income (*n*, %) ^a^			
<$45,000	368 (52)	113 (70)	255 (46)
$45,000–90,000	240 (34)	35 (22)	205 (37)
>$90,000	68 (10)	4 (2)	64 (12)
Education (*n*, %)			
Some college or less	414 (58)	113 (70)	301 (55)
Associates degree or more	297 (42)	49 (30)	248 (45)

Significance was set at *p* ≤ 0.05. ^a^ *n* = 35 answered either “don’t know” or “refused” to the income question (thus, total *n* = 676).

**Table 2 ijerph-22-00563-t002:** Linear regression of demographics and socioeconomic status to predict the perceived benefits of nature.

	Unadjusted Model			Adjusted Model ^a^		
	Estimate	95% Confidence Interval	*p*-Value	Estimate	95% Confidence Interval	*p*-Value
Race						
White	Ref.	Ref.		Ref.	Ref.	
Minoritized group	−3.82	(−6.25, −1.39)	0.002	−3.12	(−5.52, −0.71)	0.01
Age (years)						
25–34	Ref.	Ref.		Ref.	Ref.	
35–44	−2.83	(−6.05, 0.39)	0.08	−2.23	(−5.38, 0.92)	0.17
45–54	−5.21	(−8.59, −1.84)	0.002	−4.40	(−7.71, −1.08)	0.009
55–64	−4.25	(−7.33, −1.17)	0.006	−3.11	(−6.16, −0.06)	0.05
65 and over	−0.64	(−4.48, 3.20)	0.744	−0.34	(−4.11, 3.43)	0.86
Sex						
Female	Ref.	Ref.		Ref.	Ref.	
Male	−3.41	(−5.50, −1.32)	0.001	−3.85	(−5.89, −1.81)	<0.001
Education						
Associates degree or more	Ref.	Ref.		Ref.	Ref.	
Some college or less	−5.72	(−7.76, −3.68)	<0.001	−4.92	(−6.98, −2.86)	<0.001

Significance was set at *p* ≤ 0.05. ^a^ Model adjusted for race, age, sex, and education.

## Data Availability

The original contributions presented in this study are included in the article/Appendix A. Further inquiries can be directed to the corresponding author.

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
