# Peer review of "Perceived Benefits of Nature in Diverse Populations"

_ijerph, 2025, doi:10.3390/ijerph22040563_

Round 1

Reviewer 1 Report

Comments and Suggestions for Authors

Dear Authors

I have thoroughly enjoyed reading your manuscript, which is very timely and policy relevant.

You investigated perceived benefits of nature, based on demographic and socioeconomic factors among cohort members of a community-based clinical trial with an urban greening intervention.

Results showed that males, individuals from minoritised populations, and those with lower education levels perceived fewer benefits from nature. You highlight how these perceptions are shaped by socioeconomic factors and emphasises the need for accessible, multiuse green spaces in urban planning. The study calls for further research to understand and promote the benefits of nature effectively.

Introduction

The introduction effectively outlines the multifaceted benefits of green spaces, covering physiological, psychological, social, and environmental outcomes. It provides a strong rationale for the importance of understanding perceived benefits of nature.

I recommend writing out the Research Question so it is clear to the reader what the main aim of your study is (e.g., Research question: What are the perceived benefits of nature among cohort members of a community-based clinical trial with an urban greening intervention, and how do these perceptions vary based on demographic and socioeconomic factors?).

Section starting line 41: I recommend reconsidering the use of the term ‘greenness’, which to me indicates exposure to the colour ‘green’ in nature. Is there any evidence claiming that it is the green colour in nature that is beneficial? Or is it the exposure to ‘nature’ in general, meaning that you also get the benefits from exposures during the winter period where there perhaps is not much ‘greenness’. In addition, I can imagine examples where there is plenty of greenness but low biodiversity and associated well-being effects. If the term ‘greenness’ is used I recommend clarifying its meaning in the context of your study.

Line 66: “Additionally, the elderly and children also tend to see more health benefits more from green exposures”. I think there is a superfluous ‘more’ in this sentence.

To clarify key points in the introduction, the authors should consider breaking it into shorter paragraphs, each focusing on a specific type of benefit (e.g., physiological, psychological, social, environmental) to enhance readability.

Terms like “urban greening initiatives” should be briefly defined for clarity, especially for readers unfamiliar with the term. Describe briefly what the urban greening intervention entails.

Methods

The methodology is well-structured, outlining participant recruitment and data collection processes clearly.

I recommend providing a brief explanation or justification for the choice of the PBNQ used in this study.

Results

The results section presents findings clearly, with relevant statistics.

However, it is very short and I believe it could be expanded upon and explained in a bit more depth.

I recommend including tables AND figures to summarise the key findings for better visualization. Perhaps use a scatter plot to visualise the linear regression, alongside a table reporting the key regression coefficients and their statistical significance levels, providing both visual interpretation of the relationship and precise details about the model fit.

Line 150: I recommend clarifying terms such as “unadjusted” and “fully adjusted models”. A brief explanation for clarity would suffice.

I recommend the authors contextualise the findings by briefly discussing the implications of the demographic data presented. E.g., how does the demographic makeup of participants relate to their perceived benefits?

Discussion

The discussion connects findings to existing literature and theoretical frameworks effectively.

I recommend organising the discussion thematically (e.g., gender differences, socioeconomic factors, minority populations) to enhance flow and coherence and align directly with the research question and results section. This could be done effectively by using subheadings, breaking up the discussion into sections, which will guide readers through the analysis of results and show a clearer link to the research question.

The authors mention limitations; however, I recommend discussing the implications of these limitations in greater detail and within a broader context, grounding them in a real-world setting.

Conclusion

The authors have succinctly summarised the main findings and emphasised the need for further research. They should consider strengthening their conclusion by including a direct call to action emphasising the importance of addressing perceived benefits in urban policy and planning explicitly. Something along the lines of: “Future urban planning efforts must prioritise the creation of accessible, safe, and multifunctional green spaces to enhance community perceptions of nature and its benefits for everyone, particularly for those who need it most. Reinforce the significance of the study and its findings for public health and urban policy in one or two sentences.

Good luck with the publication of your paper.

Author Response

I have thoroughly enjoyed reading your manuscript, which is very timely and policy relevant.

We appreciate this compliment.

You investigated perceived benefits of nature, based on demographic and socioeconomic factors among cohort members of a community-based clinical trial with an urban greening intervention.

Yes, this summary is an accurate one.

Results showed that males, individuals from minoritised populations, and those with lower education levels perceived fewer benefits from nature. You highlight how these perceptions are shaped by socioeconomic factors and emphasises the need for accessible, multiuse green spaces in urban planning. The study calls for further research to understand and promote the benefits of nature effectively.

Yes, the above sentences accurately summarize our findings.

Introduction

The introduction effectively outlines the multifaceted benefits of green spaces, covering physiological, psychological, social, and environmental outcomes. It provides a strong rationale for the importance of understanding perceived benefits of nature.

Thank you.  We appreciate this positive comment on our introduction.

I recommend writing out the Research Question so it is clear to the reader what the main aim of your study is (e.g., Research question: What are the perceived benefits of nature among cohort members of a community-based clinical trial with an urban greening intervention, and how do these perceptions vary based on demographic and socioeconomic factors?).

Based on your suggestion, we have added two research questions at the end of the Introduction.  The newly added text reads as follows:

More specifically, we investigated two research questions: What are the perceived benefits of nature among cohort members enrolled in a community-based clinical trial with an urban greening intervention?, and How do cohort members’ perceptions vary based on demographic and socioeconomic factors?  

Section starting line 41: I recommend reconsidering the use of the term ‘greenness’, which to me indicates exposure to the colour ‘green’ in nature. Is there any evidence claiming that it is the green colour in nature that is beneficial? Or is it the exposure to ‘nature’ in general, meaning that you also get the benefits from exposures during the winter period where there perhaps is not much ‘greenness’. In addition, I can imagine examples where there is plenty of greenness but low biodiversity and associated well-being effects. If the term ‘greenness’ is used I recommend clarifying its meaning in the context of your study.

We appreciate your consideration of these areas. We retained the term greenness and now have clarified our use of greenness by defining the term when it appears in line 41.

Line 66: “Additionally, the elderly and children also tend to see more health benefits more from green exposures”. I think there is a superfluous ‘more’ in this sentence.

Thank you for the careful reading.  We have corrected this sentence.

To clarify key points in the introduction, the authors should consider breaking it into shorter paragraphs, each focusing on a specific type of benefit (e.g., physiological, psychological, social, environmental) to enhance readability.

We have divided the longest paragraph in the original draft’s Introduction into two paragraphs to enhance readability. Several of the opening paragraphs were structured based on areas of benefits, and we thought that additional shifting there might result in some paragraphs with very few sentences.

Terms like “urban greening initiatives” should be briefly defined for clarity, especially for readers unfamiliar with the term. Describe briefly what the urban greening intervention entails.

We appreciate this point and now have incorporated a definition of “urban greening initiatives” and examples when the term first appears (last paragraph of Introduction, approximately lines 84-85).

Methods

The methodology is well-structured, outlining participant recruitment and data collection processes clearly.

Thank you for the compliment on our Methods section.

I recommend providing a brief explanation or justification for the choice of the PBNQ used in this study.

Based on your suggestion, we have added a brief justification for this measure in the Methods section.

Results

The results section presents findings clearly, with relevant statistics.

Thank you for this compliment.

However, it is very short and I believe it could be expanded upon and explained in a bit more depth. I recommend including tables AND figures to summarise the key findings for better visualization. Perhaps use a scatter plot to visualise the linear regression, alongside a table reporting the key regression coefficients and their statistical significance levels, providing both visual interpretation of the relationship and precise details about the model fit.

We have added a table (Appendix 2) to provide mean sum scores of the PBNQ by demographic characteristics. Given the editorial guidelines in the instructions to authors for a brief report, we are limited to 2 tables or figures. To adhere to these limits, we chose tables that we thought best represented our data concisely.

Line 150: I recommend clarifying terms such as “unadjusted” and “fully adjusted models”. A brief explanation for clarity would suffice.

We have added information to clarify based on this suggestion and defined as race, sex, age and education.

I recommend the authors contextualise the findings by briefly discussing the implications of the demographic data presented. E.g., how does the demographic makeup of participants relate to their perceived benefits?

In the revised document, we have added information on other areas of significant findings for demographics and perceived benefits in the adjusted model, and we discuss these findings more fully in the next section of the manuscript.

Discussion

The discussion connects findings to existing literature and theoretical frameworks effectively.

Thank you for this positive comment.

I recommend organising the discussion thematically (e.g., gender differences, socioeconomic factors, minority populations) to enhance flow and coherence and align directly with the research question and results section. This could be done effectively by using subheadings, breaking up the discussion into sections, which will guide readers through the analysis of results and show a clearer link to the research question.

Based on your suggestion, we have added subheadings to guide readers through the discussion of results according to demographic and socioeconomic characteristics.  We believe that this addition accentuates the thematic discussion of the study’s results.

The authors mention limitations; however, I recommend discussing the implications of these limitations in greater detail and within a broader context, grounding them in a real-world setting.

Thank you for this suggestion.  We have added the following to the Limitations section:

Building from this work, future studies might follow cohorts in multiple and differing geographic locations across time employing an intersectional approach to investigate group differences more fully, and urban planners and policymakers should assess differential access to safe, multipurpose green spaces across population groups.

Conclusion

The authors have succinctly summarised the main findings and emphasised the need for further research. They should consider strengthening their conclusion by including a direct call to action emphasising the importance of addressing perceived benefits in urban policy and planning explicitly. Something along the lines of: “Future urban planning efforts must prioritise the creation of accessible, safe, and multifunctional green spaces to enhance community perceptions of nature and its benefits for everyone, particularly for those who need it most. Reinforce the significance of the study and its findings for public health and urban policy in one or two sentences.

Thank you for the suggestion on strengthening the language in the Conclusion.  We have used your input above and added the following:

Although additional research is needed to better understand contributors to these perceptions, future urban planning efforts should prioritize the creation and maintenance of accessible, convenient, safe, and multifunctional green spaces to enhance community perceptions of nature and its benefits for everyone, particularly for those who need it most.

Good luck with the publication of your paper.

Thank you.  We appreciate your detailed review of the manuscript and helpful comments.

Reviewer 2 Report

Comments and Suggestions for Authors

This paper explored the relationships between nature appreciation and demographics. 

The paper is related to the journal's topic. However, some of the results could be found across previous studies, which used experiments or more vigorous methods in examining the results.

Sillman, Delaney, et al. "Do sex and gender modify the association between green space and physical health? A systematic review." Environmental research 209 (2022): 112869.

Browning, Matthew HEM, and Alessandro Rigolon. "Do income, race and ethnicity, and sprawl influence the greenspace-human health link in city-level analyses? Findings from 496 cities in the United States." International journal of environmental research and public health 15.7 (2018): 1541.

Suppakittpaisarn, Pongsakorn, et al. "Durations of virtual exposure to built and natural landscapes impact self-reported stress recovery: evidence from three countries." Landscape and Ecological Engineering 19.1 (2023): 95-105.

Please state clearly how this paper contribute a novel idea to the body of literature. 

Gaps in the literature should be addressed more clearly. And how the results may apply to the bigger body of evidence should be further discussed. 

Otherwise, the paper is well structured and is methodologically sound. I'm looking forward to the published version.

Author Response

This paper explored the relationships between nature appreciation and demographics. 

Yes, this summary statement succinctly describes the study focus.

The paper is related to the journal's topic. However, some of the results could be found across previous studies, which used experiments or more vigorous methods in examining the results.

Thank you for including these references. We have included information below regarding each.

Sillman, Delaney, et al. "Do sex and gender modify the association between green space and physical health? A systematic review." Environmental research 209 (2022): 112869.

This systematic review included 62 articles that examined whether the protective associations between physical health (i.e., seven health outcomes) and green space exposure were modified by sex or gender. Overall, across most of the articles included in the review, the associations were stronger for women, especially in terms of outcomes related to obesity and mortality. In examining effect modification with stratified analyses based on distance and type of green space, compared to men, women had somewhat stronger associations between physical health and green space when the green space was very near their residence and for overall greenness (i.e., land cover). Although this article examines related areas and reviews some literature related to our study, its focus on documented associations between physical health and green space exposure differs from our focus on perceived benefits of greenness, which may influence community members’ time in nature and thus their health. Sillman et al. include a call for additional research in this and related areas, and we believe that our study contributes useful information toward better understanding the relationship between humans, including sex differences as discussed in the Sillman et al. review, and green spaces. We appreciate your including this citation, which we now include in our article.   

Browning, Matthew HEM, and Alessandro Rigolon. "Do income, race and ethnicity, and sprawl influence the greenspace-human health link in city-level analyses? Findings from 496 cities in the United States." International journal of environmental research and public health 15.7 (2018): 1541.

Like the article above, the Browning and Rigolon article focuses on the relationship between health and green space exposure. It includes assessment of the effects of various factors, including race, ethnicity, income, and urban sprawl, on this relationship. Based on analysis of 496 cities, the authors found positive associations between green spaces and health in some of their models and that the association was moderated by race and ethnicity to some degree. For example, when analyzing data from cities where non-Hispanic Black citizens are the population majority, increased tree canopy cover was associated with lower obesity. However, the authors note that their mixed overall findings, including those on the role of race and ethnicity in moderating potential effects, indicate the importance of future work in this area. Browning and Rigolon’s city-level analysis centers in connections between greenness exposure and human health. Although there are overlapping interests, such as in how race may moderate such connections, the main foci differ—Browning and Rigolon examine associations between green space and health whereas our study examines demographic and socioeconomic characteristics and the perceived benefits of nature. Though these areas are related, as perceptions of nature’s benefits may influence time spent in nature which may, in turn, influence health, they are distinct areas of inquiry. Our study begins to address Browning and Rigolon’s call for more inquiry into the potential moderating effects of race and ethnicity in greenness studies. Thank you for suggesting this reference. We have cited it in our revised article.

Suppakittpaisarn, Pongsakorn, et al. "Durations of virtual exposure to built and natural landscapes impact self-reported stress recovery: evidence from three countries." Landscape and Ecological Engineering 19.1 (2023): 95-105.

Suppakittpaisarn et al.’s research explores how virtual exposure to urban and natural settings may influence recovery from stress. In this experimental study employing a sample spanning three countries, the authors tested time durations for virtual (i.e., video) exposures. The findings of their analysis revealed that, compared to exposure to virtual urban environments, exposure to virtual natural environments promoted stress recovery for women but not for men. Thank you for sharing this reference. We agree that it addresses an interesting area of the human-nature interface and is useful for understanding recovery from stress and related areas. Given that our article is a brief report with space limitations, we chose not to focus on virtual, pictorial, and similar areas of work; thus, we have not added this citation to the revised manuscript. It will be useful for our continued work in this area. 

Please state clearly how this paper contribute a novel idea to the body of literature. 

In the revised document, we have elaborated on the gap in the academic literature that our study addresses and the contributions of this study as well as clarified potential applications of the study’s results. The revisions include several changes to the last two paragraphs of the Introduction. For example, in this section, we now emphasize the study goals, expand on terminology used, directly state the study’s hypotheses, and explain how the findings may be applied. 

Gaps in the literature should be addressed more clearly. And how the results may apply to the bigger body of evidence should be further discussed. 

Please see our response to the suggestion just above.

Otherwise, the paper is well structured and is methodologically sound. I'm looking forward to the published version.

Thank you for the compliments on the manuscript’s structure and methods and your comments to help us strengthen the manuscript.

Reviewer 3 Report

Comments and Suggestions for Authors

Dear author(s),

Below are some comments for your consideration to improve your manuscript:

  1. Please clearly highlight the research gap in the introduction to strengthen the rationale for your study.

  2. Clarify the difference between p < 0.001 and p = 0.001 in lines 153 and 156 to ensure consistency and accuracy in statistical reporting.

  3. The results are too general. I suggest incorporating the PBNQ results into the manuscript to provide a more detailed analysis of differences across gender, age, and other socio demographic factors. Consider using ANOVA or a similar statistical test to enhance the manuscript's depth and rigor.

Comments on the Quality of English Language

Please enhance the academic language for greater clarity and precision. Overall, the manuscript demonstrates good proficiency in English.

Author Response

Below are some comments for your consideration to improve your manuscript:

  1. Please clearly highlight the research gap in the introduction to strengthen the rationale for your study.

Based on your suggestion, we have more clearly highlighted the gap in the academic literature that our study addresses. This revision better positions our study and accentuates its purpose. As suggested, these changes were made to the last sections of the Introduction.

  1. Clarify the difference between p < 0.001 and p = 0.001 in lines 153 and 156 to ensure consistency and accuracy in statistical reporting.

Thank you for your careful read of the manuscript. We have checked these uses to confirm reporting accuracy for the unadjusted model. In this case, the reporting is correct—one p value equaled 0.001 (i.e., male sex), and the other was less than 0.001 (i.e., less education).

  1. The results are too general. I suggest incorporating the PBNQ results into the manuscript to provide a more detailed analysis of differences across gender, age, and other socio demographic factors. Consider using ANOVA or a similar statistical test to enhance the manuscript's depth and rigor.

We appreciate this suggestion; however, because the manuscript is a Brief Report, we considered options to streamline the number of tables and reporting of general results to maintain the focus on the study’s central areas and keep the information succinct to meet expectations for this type of short report. We have added a table (Appendix 2) of the mean PBNQ scores by sociodemographic characteristics and discussed the results in the manuscript to increase the depth and rigor in relation to this style of manuscript.

Comments on the Quality of English Language

Please enhance the academic language for greater clarity and precision. Overall, the manuscript demonstrates good proficiency in English.

Thank you for the positive comment on our overall use of the English language. We have revised the language in several places in the manuscript to enhance clarity and precision.

Round 2

Reviewer 3 Report

Comments and Suggestions for Authors

Dear Author,

Here are some recommendations for improving your manuscript:

  1. In line 14, consider changing the word "men" to "sex/gender" for better inclusivity and clarity.
  2. It is suggested that the author include a figure illustrating the research design to help readers better understand the research flow.

Author Response

Here are some recommendations for improving your manuscript:

  1. In line 14, consider changing the word "men" to "sex/gender" for better inclusivity and clarity.

We appreciate your points regarding inclusivity and work to use an inclusive lens in our research. The sentence in line 14 used men as an example of a group that has been shown in prior studies to benefit more from nature exposure. However, because our use of this example raised concerns regarding inclusivity, we have deleted it. The abstract’s opening sentence conveys its points well without this example.

  1. It is suggested that the author include a figure illustrating the research design to help readers better understand the research flow.

Based on your suggestion, we have added a figure illustrating the research flow of our study. We reference this figure (i.e., Supplementary Figure 1) at the end of the first paragraph of the Methods section.

Thank you for your review of our work.